# Different Eating Habits Are Observed in Overweight and Obese Children Than in Normal-Weight Peers

**DOI:** 10.3390/children11070834

**Published:** 2024-07-09

**Authors:** Żaneta Malczyk, Agnieszka Pasztak-Opiłka, Agnieszka Zachurzok

**Affiliations:** 1Department of Pediatrics, Faculty of Medical Sciences in Zabrze, Medical University of Silesia in Katowice, 41-800 Zabrze, Poland; agnieszkazachurzok@poczta.onet.pl; 2Centrum Navigare, 41-300 Dąbrowa Górnicza, Poland; centrum@navigare.edu.pl

**Keywords:** children, eating behavior, CEBQ, obesity

## Abstract

Background: Obesity is diagnosed in 13.6% of early primary school children in Poland. Its presence at this age increases the risk of obesity occurrence in adulthood. Therefore, it is important to properly shape eating behaviors at the stage of childhood and identify incorrect eating styles. Methods: This study aimed to investigate whether overweight and obese children differ significantly from children with normal body weights in terms of their eating styles. For the materials and methods, 43 mothers of overweight or obese children aged 3–10 years and 88 mothers of normal-weight children aged 3–10 years completed a questionnaire related to sociodemographic factors and the Children’s Eating Behaviour Questionnaire. Results: The overweight and obese children, compared with normal-weight children, scored higher on the food responsiveness (*p* = 0.009) and emotional overeating (*p* = 0.013) scales and lower on the satiety responsiveness (*p* = 0.025) and slowness in eating scales (*p* < 0.0001). No significant difference was found for other subscales between the studied groups. In the group of overweight and obese children, the child’s age correlated negatively with enjoyment of food, as did the mother’s BMI with slowness in eating. Conclusions: The results indicate the presence of significant differences in eating styles between normal-weight children and overweight or obese children. Identifying families at high risk of inappropriate eating behaviors and educating them appropriately can reduce the risk of children becoming overweight or obese.

## 1. Introduction

Obesity is increasingly being diagnosed in the pediatric population [1]. Based on the WHO European Childhood Obesity Surveillance Initiative (COSI) report, 32.3% and 13.6% of Polish children between the ages of 7 and 9 years can be diagnosed as overweight and obese, respectively (https://www.who.int/europe/publications/i/item/WHO-EURO-2022-6594-46360-67071 (8 November 2022)). Unfortunately, the problem of excessive body weight in children, especially at an extremely young age, is often underestimated and seen as a temporary period [2]. Adequate nutrition of preschool and school-age children determines their optimal physical, mental, and social development [2]. Obesity during childhood and adolescence is related to an increased risk of serious health consequences such as cardiovascular disease, high blood pressure, diabetes, obstructive sleep apnea, hypertension, hepatic steatosis, and polycystic ovary syndrome. Obesity is also a risk factor for developing many types of cancer [3]. Obese children experience a number of psychosocial problems that significantly affect their quality of life and wellbeing. Children who are obese or overweight have low self-esteem and are more likely to have depression, anxiety, or a negative body image. Obesity also has a negative impact on the emotional development of children. Children with excessive weight are more likely to be the victims of discrimination, social isolation, and bullying. Due to the complications of obesity, they miss school more frequently, thereby affecting their school performance negatively [3].

Childhood is a critical period for the development and maintenance of obesity into adulthood [4]. The body fat of overweight infants increases during the first year of life and then slowly decreases and increases again around the age of six. Obesity rebounds occur earlier in infants with higher amounts of adipose tissue. The earlier the return to obesity, the higher the risk of being overweight as an adult [4]. Risk factors for an early adiposity rebound are preterm birth, a low birth weight, modified milk feeding, and early introduction of sugar-sweetened foods [4].

It has been shown that childhood obesity significantly increases the risk of being obese as an adult [5,6]. Ward et al. estimated the probability that an obese 2 year-old child would still be obese at the age of 35 years to be almost 75%, while for a 19 year-old, it was more than 88% [5].

In light of the increasing prevalence of overweight and obese children in the pediatric population, there is growing interest in the eating behaviors already exhibited by the youngest children. Research shows that childhood eating habits can continue into adulthood [7]. Numerous studies indicate a correlation between eating behavior and body weight [8,9,10,11]. Early childhood is the time when eating behaviors are shaped, and they depend mainly on the parents [10]. In the fourth year of life, children develop their own eating habits [12]. Inappropriate eating behaviors which occur in childhood can result in an abnormal body weight in adulthood. Dubois et al. showed that children who are picky eaters are twice as likely to be underweight, while overeating increases the risk of being overweight by six times [13]. Marchi and Cohen found that pickiness in eating increases the risk of developing an eating disorder during adolescence [14]. Viena et al. reported that children with low body weights have the lowest positive responses to food [15]. According to Webber, children who are underweight have worse appetites than slim children with normal body weights [16].

Overweight and obese children have a stronger response for the smell, flavor, or presentation of the food. They experience more pleasure in eating and tend to eat more rapidly and drink a lot of sugary beverages, but their feeling of satiety is lower than that in children with normal body weights [8,9,10,11].

Overeating may be associated with compulsive eating and becoming overweight [17,18,19]. Therefore, to prevent becoming overweight or obese, it is important to develop appropriate habits in children and to identify behaviors that contribute to overeating early.

There are few studies examining the impact of the eating style of Polish children on the incidence of obesity. Therefore, the aim of this study was to investigate whether overweight and obese children differ significantly in terms of their eating styles compared with children with normal body weights [18].

## 2. Material and Methods

The study is a continuation of the published research by Malczyk et al. on validation of the Children’s Eating Behaviour Questionnaire (CEBQ) in the Polish population [20]. The study group consisted of 43 mothers aged between 24 and 49 years (M = 35.84, SD = 5.93) of overweight or obese children in the age range of 3–10 years (M = 7.14, SD = 2.45; 21 girls, 22 boys). According to this small number of patients, the overweight and obese children’s data were analyzed jointly.

The control group consisted of 88 mothers aged between 21 and 47 years (M = 33.99, SD = 5.32) of normal-weight children age-matched to the study group (M = 6.75, SD = 2.28; 52 girls, 36 boys). The survey procedure involved the mothers completing the self-prepared questionnaire related to sociodemographic factors and the CEBQ adapted to Polish conditions using the paper-and-pencil method in a non-limited time [20]. The CEBQ is a tool for assessing nutritional behavior in children developed by Wardle et al. in 2001. It consists of 35 questions completed by the parent, containing eight subscales of eating styles: four food-approaching subscales (food responsiveness, emotional overeating, enjoyment of food, and desire to drink) and four food-avoidant subscales (emotional undereating, satiety responsiveness, slowness in eating, and fussiness). The Polish version of the CEBQ, validated by Malczyk et al., has high reliability (Cronbach’s alpha coefficient = 0.78) and validity (Kaiser–Mayer–Olkin (KMO) coefficient >0.5) [20].

The mothers rated their children’s eating behavior on a five-point Likert scale (1 = never; 2 = rarely; 3 = sometimes; 4 = often; 5 = always). Five statements had reverse scoring. They also completed a questionnaire related to sociodemographic factors. In the study groups, the heights and weights of the mothers and children were measured, the BMI was calculated for each of them, and the BMI z scores for the children were calculated. The BMI-z score was calculated according to the recommendations of the World Health Organization. Age and gender were taken into account in the calculations [21].

This study was conducted according to the Declaration of Helsinki and approved by the local ethics committee (approval no. PCN/CBN/0022/KB1/64/I/20/21).

Data were compared using Statistica 14.0 EN software. For all the parameters tested, the conformity of their distributions with the normal distribution was checked using the Shapiro–Wilk test. The homogeneity of variance was evaluated with Levene’s test. For variables with a normal distribution, the significance of differences between groups was checked using a *t*-test. For the variables not distributed normally, the non-parametric Mann–Whitney U test was applied.

## 3. Results

Descriptive statistics for the mothers’ and children’s ages and the mothers’ and children’s BMIs are presented in Table 1. The BMI was significantly higher in the mothers of children with excessive weight compared with the mothers of normal-weight children (*p* = 0.0002). The sociodemographic description of the mothers and families is shown in Table 2. 

The mean, median, and standard deviation values for each of the CEBQ subscales are reported in Table 3. The overweight and obese children, compared with the normal-weight children, scored higher on the food responsiveness and emotional overeating scales and lower on the satiety responsiveness and slowness in eating scales. No significant difference was found for the desire to drink, emotional undereating, fussiness, or enjoyment of food scales between the study groups.

In the group of overweight and obese children, a significant correlation was found between the mother’s BMI and slowness in eating (r = −0.31, *p* = 0.04). In addition, food responsiveness (r = 0.54, *p* = 0.0001) and enjoyment of food (r = 0.41, *p* = 0.006) correlated positively (and satiety responsiveness negatively (r = −0.53, *p* = 0.0003)) with the child’s age. However, no relationship was found between the child’s BMI z score and the CEBQ subscales. The mother’s education level correlated negatively with food responsiveness (r_γ_ = −0.28, *p* = 0.03) and enjoyment of food (r_γ_ = −0.31, *p* = 0.02) and positively with the desire to drink (r_γ_ = 0.28, *p* = 0.04). The number of children in the family correlated negatively with achieving satiety (r_γ_ = −0.32, *p* = 0.02).

Interestingly, a different relationship was observed in the group of children without excessive body weight. In their case, the mother’s BMI did not correlate with any subscale of the CEBQ, and the child’s age correlated negatively only with enjoyment of food (r = −0.25, *p* = 0.02). A negative relationship was observed between the BMI z score and slowness in eating (r = −0.26, *p* = 0.02), a positive one was found between education level and emotional overeating (r_γ_ = 0.26, *p* = 0.007), and a negative one was found between the number of children in the family and emotional undereating (r_γ_ = −0.23, *p* = 0.01).

## 4. Discussion

Numerous reports in the literature indicate that childhood eating behaviors continue into adulthood. In light of the increasing number of people with excessive body weight, it seems extremely important to identify and modify inappropriate eating habits as early as possible. Therefore, the aim of this study was to verify whether there are different eating behaviors in overweight and obese children compared with normal-weight children. The authors showed significant differences in eating styles between the study groups. The children with excessive weight scored higher on the food responsiveness and emotional overeating scales and lower on the satiety responsiveness and slowness in eating scales. In the studied groups, different correlations were also found between the eating styles and selected interview data, such as the mother’s BMI, education level, child’s age, and number of children in the family.

We found that the overweight and obese children scored higher on the food responsiveness scale compared with their normal-weight peers. Food responsiveness is defined as an increased interest in food and a heightened response to food cues, such as the appearance of food or its color, smell, or taste [22]. Children scoring high on this subscale are characterized by constantly thinking about food and demanding the next meal regardless of how satiated they are. French’s work showed that overly positive responsiveness to external food stimuli is associated with excessive food intake and an increased risk of future abnormal body weights [22]. Similar correlations were also observed by Pasoss [23] and Sleddens et al. [24]. In our group of obese children, a negative correlation was observed between the mother’s education and the child’s food responsiveness. This may be due to the greater knowledge of educated mothers about the principles of healthy eating and the need to form good eating habits in children. Inappropriate family dietary patterns may be imitated by children, especially during adolescence, when the quantity and quality of the products consumed increasingly depend on them. These observations are in line with the report by Vian et al. [15].

This study showed a significant difference between the studied groups in terms of emotional overeating. Interestingly, in the group of normal-weight children, a positive correlation was found between the mother’s education and emotional overeating, whereas this relationship was not observed in the overweight children. The main determinant of eating in children with high scores on this subscale is the feeling of emotional tension. Experiencing sadness, anger, or another strongly marked emotion leads to eating more than usual. Research indicates that obese people are more likely to have certain temperamental traits such as pessimism, poorer stress coping, timidity, and low openness to new experiences [25]. Numerous studies linking emotions and excessive food intake can be found in the literature. Strien et al. suggested that emotional eating has its basis in underdeveloped stress and emotion regulation strategies [26]. These observations were also confirmed by Jáuregui-Lobera et al. [27]. Individuals who tend to eat in response to negative emotions such as anxiety or irritability have been observed to have an increased risk of being overweight [26]. In children, emotional overeating may also result from an inappropriate approach to nutrition by caregivers [26]. Food is often treated by parents as a form of reward or consolation. This causes children to develop inappropriate ways of coping with stress and negative feelings [26]. Sometimes, caregivers make their good moods dependent on the child eating, thus causing them to associate the act of eating with feeling positive emotions.

In their work, Jáuregui-Lobera and Montes-Martínez mentioned an important mechanism concerning children: eating to please someone rather than to satisfy hunger [27]. This may explain the positive correlation observed in the control group between maternal education and emotional overeating. Mothers with higher education may also be more demanding toward their children in the area of food.

In response to negative emotions or stress in some children, emotional undereating may be observed. This may be due to individual reactions to stress. For the emotional undereating scale, no significant difference was found in this study. These results are consistent with the findings of Webber et al. [16].

Satiety responsiveness is associated with behaviors such as finishing a meal early, undereating, and leaving part of the portion on the plate when the fullness threshold is reached. The correct response to the feeling of satiety determines the correct regulation of energy intake. Parents imposing the time of eating, type of food, and portion size may result in the child misreading hunger and satiety signals [26,27,28]. Excessive use of orders or incentives by parents causes the child to start eating when the parent thinks it is right, losing the natural ability to self-regulate eating [15]. Eating in the absence of of hunger signals often results from a tendency to derive pleasure from the act of eating itself [22]. Carnell et al. showed that children who were more likely to reach for a snack after a heavy meal had higher scores on the food responsiveness and enjoyment of food scales and lower scores on the satiety responsiveness scale [29]. This explains the lower scores obtained in this study on the satiety responsiveness scale for overweight and obese children compared with normal-weight children.

Research indicates that infants are quite sensitive to internal hunger and satiety signals. As children grow older, they gradually lose the ability to effectively self-regulate energy intake, which promotes overeating and weight gain [29]. The authors of the present study also showed a negative correlation between the age of the child and satiety responsiveness. The number of children in the family correlated negatively with satiety responsiveness in children with excessive weight. This may be due to the reduced ability to focus attention on the child and the lack of early recognition of satiety signals already manifested by the youngest children. The economic aspect was also not insignificant; a lower income may influence eating behaviors, such as the choice of inferior quality products, more frequent recourse to processed foods and ready-made, quick-to-serve dishes, although these hypotheses certainly require further analysis.

In our study, the overweight and obese children had faster rates of eating than the normal-weight children, and slowness in eating correlated negatively with the maternal BMI in the study group and BMI z score in the control group. Reports in the literature show that fast eating, despite consuming more calories, does not result in a more rapid onset of satiety [30]. Barkeling et al. compared the lunch consumption times of 11 year-old obese and normal-weight children, finding that the obese children ate faster and, in contrast to the normal-weight children, did not slow down at the end of the meal [30]. Many factors, such as genetic factors and temperament, can influence the rate of eating. The existing negative relationship between slowness in eating and activity and sociability indicates that the higher the motor activity, the lower the tendency to eat slowly, and the higher the sociability, the higher the rate of eating [20]. Environmental factors also influence the rate of eating. In adults living hectic lives, a clear link was found between fast food consumption and excessive body weight and eating disorders [31]. Similar relationships were observed in the pediatric population [23,24].

Enjoyment of food is defined as feeling the joy associated with eating and looking forward to the next opportunity to eat [22]. In the present study, there were no differences in the response to food among the studied groups, but different correlations were found. In children with excessive weight, a positive correlation was observed between the child’s age and enjoyment of food. Older children showed a greater willingness to reach for more meals, which they associated with feeling pleasure. An inverse relationship was observed in the control group. Perhaps normal-weight children do not experience as much pleasure from the act of eating itself as their obese peers. Moreover, enjoyment of food in the study group, like food responsiveness, correlated negatively with the mother’s education, confirming previous observations of their greater awareness of healthy eating.

Desire to drink describes the child’s behavior of wanting to consume liquids and frequently asking for something to drink. In recent years, there has been an increase in publications on the consumption of sugary drinks, both among adults and children. In some people, the excess energy provided by sugary drinks is crucial in becoming overweight or obese [10]. Sweetman and Wardle found that people who consume sugary drinks show similar food choice preferences and are more likely to be overweight [32]. The authors found no significant difference in the desire to drink among the children studied, but in the group of overweight and obese children, they found a positive correlation between the mother’s level of education and the desire to drink. It may be that mothers with higher education are more attentive to children demanding too many drinks. Nevertheless, this hypothesis certainly requires further research.

Fussiness is the tendency to have food preferences and refusing to try new flavors. This eating style is usually associated with being underweight. According to Dubois, children expressing fussiness are twice as likely to be underweight compared with children without this problem [13]. In 1990, Marchi and Cohen already proved the relationship between selective eating and the occurrence of anorexia nervosa [14]. Weber et al. showed that obese children have lower levels of fussiness compared with children with normal body weights [16]. In our study, fussiness was not related to the BMI z score.

Our study presents some limitations. The main one is the limited number of patients, which made it impossible to analyze the overweight and obese children separately. The second one is the lack of prospective observation. The correlation between the eating style and BMI should be verified in longitudinal studies.

## 5. Conclusions

Our results confirm the feasibility of using the CEBQ adapted to Polish conditions to assess eating behaviors in children. The results indicate the presence of significant differences in eating styles between normal-weight children and overweight or obese children. They are characterized by an increased interest in food and excessive eating, which might be a strategy for coping with stress and emotions, as well as with poor satiety signal responsiveness and faster eating. This indicates that their eating styles may be related to a higher risk of excessive calorie intake, leading to becoming overweight or obese. Identifying families at high risk of inappropriate eating behaviors and educating them appropriately can reduce the risk of children becoming overweight or obese.

## Figures and Tables

**Table 1 children-11-00834-t001:** Descriptive statistics for variables: mother’s age, child’s age, mother’s BMI, and child’s BMI z scores.

	Children with Normal Weights (*n* = 88)		Overweight or Obese Children (*n* = 43)		*p*
	Mean	SD	Mean	SD	
**Mother’s age (years)**	33.99	5.32	35.84	5.93	0.1
**Child’s age (years)**	6.75	2.28	7.14	2.45	0.3
**Mother’s BMI (kg/m^2^)**	23.65	3.58	26.90	5.02	0.0002
**Child’s BMI z score**	−0.23	0.86	1.82	0.45	<0.0001

**Table 2 children-11-00834-t002:** Sociodemographic description of mothers.

	Children with Normal Weight (*n* = 88)	Overweight or Obese Children (*n* = 43)
**Place of living**		
Village (%)	12.5	7
City (%)	87.5	93
**Number of children in the family**		
One (%)	31	21
Two (%)	50	46.5
Three (%)	11	25.5
Four (%)	3.5	7
Five or more (%)	4.5	0
**Family status**		
Complete family (%)	90	81
Incomplete family (%)	10	19
**Mother’s education level**		
Tertiary education (%)	47.8	34.9
Secondary education (%)	31.8	39.5
Vocational education (%)	13.6	18.6
Primary education (%)	6.8	7.0

**Table 3 children-11-00834-t003:** CEBQ subscale characteristics in mothers of normal-weight children and overweight or obese children.

	Children with Normal Weight (*n* = 88)			Overweight or Obese Children (*n* = 43)			*p*
	Mean	Median	SD	Mean	Median	SD	
**Food responsiveness**	11.71	11.0	4.68	13.93	13.0	4.97	**0.009**
**Emotional undereating**	10.52	10.0	3.90	10.46	10.0	3.77	0.95
**Fussiness**	15.66	16.0	5.07	14.32	15.0	4.76	0.15
**Desire to drink**	8.27	8.0	3.14	8.23	8.0	2.32	0.85
**Emotional overeating**	7.58	7.0	2.49	9.781	9.0	4.47	**0.013**
**Satiety responsiveness**	14.95	15.0	3.46	13.56	14.0	2.98	**0.025**
**Slowness in eating**	12.64	12.0	3.08	10.25	10.0	2.98	**<0.0001**
**Enjoyment of food**	12.79	13.0	3.76	14.12	14.0	3.36	0.05

## Data Availability

The data presented in this study are available on request from the corresponding author due to ethical reasons.

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
