# Peer review of "Different Eating Habits Are Observed in Overweight and Obese Children Than in Normal-Weight Peers"

_children, 2024, doi:10.3390/children11070834_

Round 1
Reviewer 1 Report
Comments and Suggestions for Authors
The study described in this manuscript addressed a well-known concern for overweight and obesity in young children. Differences in eating behaviors between overweight and obese youth versus normal weight youth have been detected previously. The manuscript includes citations that are mostly 10-15 years old and the most recent citation is from 2020. Updated background literature to support this study, particularly focused on the importance for youth in Poland, is needed to enhance the significance of the study.
An uneven sample size for comparison between the two study groups was noted. Some explanation of the uneven samples is needed.
The abstract is well-organized, but has some discrepancy regarding the correlation of the children's BMI z-scores (versus the mother's BMI, which is reported in the narrative of the manuscript) in the overweight and obese group with Slowness of Eating.
Page 1, Lines 44-45 includes a one sentence paragraph that indicates infancy and adolescence as critical periods for the development of overweight and obesity. The paragraph needs more information on these critical periods. The following paragraphs include some details about adolescence but not infancy. Since the study addresses youth that are 3 to 10 years of age, the risks specific to these ages should be addressed.
Pg. 2, L 69: remove the word "its" before the word "eating."
In methods, please describe how child BMI z-scores were calculated and the inclusion of age and gender in the calculation. Include a citation for the calculation that was used. Include the number of male and female children in each group.
In Table 2, please check the percentages for the place of living for children with normal weight. It looks that commas are used with the percentage.
Page 3, Line 104, rather than state "descriptive characteristics," which is similar terminology used when reporting sociodemographics, state that the mean, medium and standard deviation values for each of the CEBQ subscales are reported. In the methods section, reliability and validity information for the Child Eating Behavior Questionnaire is needed with an appropriate citation.
The discussion needs to be reviewed to be consistent with describing the study findings regarding Slowness of Eating.
Author Response
Dear Reviewer,
Thank You for your valuable comments and feedback provided to our article which will allow to present even better quality paper. We as authors, carefully considered the comments and responded to each of them. Here are the point-by-point answers:
- The manuscript includes citations that are mostly 10-15 years old and the most recent citation is from 2020. Updated background literature to support this study, particularly focused on the importance for youth in Poland, is needed to enhance the significance of the study.
We have updated the basic literature and included information on the epidemiology of overweight and obesity in Polish children in the introduction section (in red)
- An uneven sample size for comparison between the two study groups was noted. Some explanation of the uneven samples is needed.
Our project aimed to achieve two goals: 1. Validation of the CEBQ for Polish children - results published [PMID: 36432467], 2. assessment of eating behaviors in children with excessive body weight - current manuscript. The control group was part of a larger group that was used to validate the questionnaire, and the study group was assessed for the second part of the project. Hence the disproportion in group size. Information that the current manuscript is part of a larger project has been added in materials and methods (in red).
- The abstract is well-organized, but has some discrepancy regarding the correlation of the children's BMI z-scores (versus the mother's BMI, which is reported in the narrative of the manuscript) in the overweight and obese group with Slowness of Eating.
The correlation in abstract is corrected - we found a correlation between the mother's BMI and Slowness in Eating
- Page 1, Lines 44-45 includes a one sentence paragraph that indicates infancy and adolescence as critical periods for the development of overweight and obesity. The paragraph needs more information on these critical periods. The following paragraphs include some details about adolescence but not infancy. Since the study addresses youth that are 3 to 10 years of age, the risks specific to these ages should be addressed.
We have expanded information on the importance of excess body fat in infants and the risk factors for "obesity rebound" in Introduction (in red)
- 2, L 69: remove the word "its" before the word "eating."
Done
- In methods, please describe how child BMI z-scores were calculated and the inclusion of age and gender in the calculation. Include a citation for the calculation that was used. Include the number of male and female children in each group.
In the material and methods section, we described how child BMI z-scores were calculated and we included a citation for this calculation. Additionally, we included the number of male and female children in studied group.
- In Table 2, please check the percentages for the place of living for children with normal weight. It looks that commas are used with the percentage.
We verified data given in Table 2 and we removed commas
- Page 3, Line 104, rather than state "descriptive characteristics," which is similar terminology used when reporting sociodemographics, state that the mean, medium and standard deviation values for each of the CEBQ subscales are reported. In the methods section, reliability and validity information for the Child Eating Behavior Questionnaire is needed with an appropriate citation.
We changed term "descriptive characteristics" to "mean, median and standard deviation". Moreover, in the methods section we added information about reliability and validity for the Polish version of Child Eating Behavior Questionnaire and we added appropriate citation (in red)
- The discussion needs to be reviewed to be consistent with describing the study findings regarding Slowness of Eating.
Reviewer 2 Report
Comments and Suggestions for Authors
Dear Authors,
Thanks for the study which aims to assess the differences in eating habits between children with normal weight and children with overweight and obesity.
Manuscript needs some revisions:
1. The introduction should be improved with the studies conducted so far on children's eating habits (both on children in general and on overweight and obese children), because the authors put more emphasis on the consequences - diseases, overweight, obesity. A review of studies on children's eating habits is lacking.
2. I would suggest moving the paragraph about children eating behaviour questionnaire (CEBQ) (Lines – 60-71) to the method section.
3. In my opinion, it would be important to indicate how many children were obese in the study group children with overweight or obesity (n=43), because it significantly affects the results of the study.
4. In the discussion part, 6 out of 8 subscales of the CEBQ are analyzed. I would recommend to evaluate all subscales in the discussion, because the following subscale is missing: Emotional undereating and Fussiness.
5. Whether overweight and obese children were evaluated separately in the study, as they are indicated separately in same places (Line 150 and Line 160) in the discussion. If this is the case, it would be recommended to present separate children with overweight and children with obesity in the results presentation (Tables). If this is not the case, please specify the correct one in the discussion so as not to confuse.
6. The study does not state the limitations of the study.
Author Response
Dear Reviewer,
Thank You for your valuable comments and feedback provided to our article which will allow to
present even better quality paper. We as authors, carefully considered the comments and
responded to each of them. Here are the point-by-point answers:
Manuscript needs some revisions:
1. The introduction should be improved with the studies conducted so far on children's eating
habits (both on children in general and on overweight and obese children), because the
authors put more emphasis on the consequences - diseases, overweight, obesity. A review of
studies on children's eating habits is lacking.
Thank you for this comment, in the introduction we added information about eating
behaviour on children in general and on overweight and obese children (in red)
2. I would suggest moving the paragraph about children eating behaviour questionnaire (CEBQ)
(Lines – 60-71) to the method section.
As you suggested, we moved and rephrased the paragraph about children's eating behaviour
questionnaire (CEBQ) to the method section (in red)
3. In my opinion, it would be important to indicate how many children were obese in the study
group children with overweight or obesity (n=43), because it significantly affects the results
of the study.
Due to a small number of patients, it was impossible to analyze overweight and obese
children separately. We clarify it in the text (Material and methods) as well (in red)
4. In the discussion part, 6 out of 8 subscales of the CEBQ are analyzed. I would recommend to
evaluate all subscales in the discussion, because the following subscale is missing: Emotional
undereating and Fussiness.
In the discussion we add paragraphs regarding analysis two missing scales Emotional
undereating and Fussiness (in red)
5. Whether overweight and obese children were evaluated separately in the study, as they are
indicated separately in same places (Line 150 and Line 160) in the discussion. If this is the
case, it would be recommended to present separate children with overweight and children
with obesity in the results presentation (Tables). If this is not the case, please specify the
correct one in the discussion so as not to confuse.
As we mentioned above, due to a small number of patients, it was impossible to analyze
overweight and obese children separately. We clarify it in the text.
6. The study does not state the limitations of the study.
The limitations of the study were added.
Round 2
Reviewer 1 Report
Comments and Suggestions for Authors
Thank you for submitting responses to the initial reviewer's comments and for adequately addressing the suggestions for revising the manuscript in a timely manner.
This reviewer does not have any further major edits, except he following:
In the introduction, page 1, see line 30. Please write out the full name of COSI and place COSI in parentheses. The spelling of the word "report" is incorrect. Please correct it. In page 2, line 72, correct "the" feeling of satiety; remove the word "they" before "feeling of satiety."
Author Response
Dear Reviewer,
Thank You for your valuable comments. Changes we made in the manuscript are highlighted in yellow. Here are the point-by-point answers:
- In the introduction, page 1, see line 30. Please write out the full name of COSI and place COSI in parentheses.
We wrote the full name COSI - The WHO European Childhood Obesity Surveillance Initiative (COSI)
- . The spelling of the word "report" is incorrect.
Done
- Please correct it. In page 2, line 72, correct "the" feeling of satiety; remove the word "they" before "feeling of satiety."
Done
Kind regards
Żaneta Malczyk